# The Synergistic Anti-Tumor Activity of EZH2 Inhibitor SHR2554 and HDAC Inhibitor Chidamide through ORC1 Reduction of DNA Replication Process in Diffuse Large B Cell Lymphoma

**DOI:** 10.3390/cancers13174249

**Published:** 2021-08-24

**Authors:** Xing Wang, Dedao Wang, Ning Ding, Lan Mi, Hui Yu, Meng Wu, Feier Feng, Luni Hu, Yime Zhang, Chao Zhong, Yingying Ye, Jiao Li, Wei Fang, Yunfei Shi, Lijuan Deng, Zhitao Ying, Yuqin Song, Jun Zhu

**Affiliations:** 1Key Laboratory of Carcinogenesis and Translational Research (Ministry of Education), Department of Lymphoma, Peking University Cancer Hospital & Institute, No. 52 Fucheng Road, Haidian District, Beijing 100142, China; 1511110402@bjmu.edu.cn (X.W.); 1911210644@pku.edu.cn (D.W.); ningding@bjmu.edu.cn (N.D.); milan@bjmu.edu.cn (L.M.); yuhuidoc@bjmu.edu.cn (H.Y.); lxywumeng@bjmu.edu.cn (M.W.); feierfeng@bjmu.edu.cn (F.F.); 1811110577@bjmu.edu.cn (Y.Y.); lja03081@btch.edu.cn (J.L.); fangwei@pku.org.cn (W.F.); denglj@bjmu.edu.cn (L.D.); yingzhitao@bjmu.edu.cn (Z.Y.); 2Institute of Systems Biomedicine, Department of Immunology, Beijing Key Laboratory of Tumor Systems Biology, Peking University Health Science Center, Beijing 100191, China; huluni@hsc.pku.edu.cn (L.H.); 1711110047@bjmu.edu.cn (Y.Z.); zhongc@hsc.pku.edu.cn (C.Z.); 3Key Laboratory of Carcinogenesis and Translational Research (Ministry of Education), Department of Pathology, Peking University Cancer Hospital & Institute, No. 52 Fucheng Road, Haidian District, Beijing 100142, China; shiyunfei@bjmu.edu.cn

**Keywords:** diffuse large B-cell lymphoma, EZH2 inhibitor, synergistic effects, DNA replication process

## Abstract

**Simple Summary:**

The EZH2-targeted drugs have demonstrated notable therapeutic effects in EZH2 mutant B-cell lymphoma patients. In this study, we demonstrated that the combination of EZH2 inhibitor SHR2554 and HDAC inhibitor HBI8000 exert synergistic anti-proliferative activity in both EZH2 wide-type and mutation B-cell lymphoma. More importantly, gene expression profile analysis revealed simultaneous treatment with these agents led to dramatic inhibition of DNA replication initiator protein ORC1, which might contribute to great efficacy of combination strategy. The combination of EZH2 inhibitor and HDAC inhibitor could provide a potential therapeutic treatment for both EZH2 wide-type and mutation B-cell lymphoma patients.

**Abstract:**

Background: Upregulation of H3K27me3 induced by EZH2 overexpression or somatic heterozygous mutations were implicated in lymphomagenesis. It has been demonstrated that several EZH2-target agents have notable therapeutic effects in EZH2-mutant B-cell lymphoma patients. Here we present a novel highly selective EZH2 inhibitor SHR2554 and possible combination strategy in diffuse large B-cell lymphoma (DLBCL). Methods: Cell proliferation, cell cycle and apoptosis were analyzed by CellTiter-Glo Luminescent Cell Viability Assay and flow cytometry. Western Blot was used to detect the expression of related proteins. The gene expression profiling post combination treatment was analyzed by RNA-Seq. Finally, CDX and PDX models were used to evaluate the synergistic anti-tumor effects of the combination treatment in vivo. Results: The novel EZH2 inhibitor SHR2554 inhibited proliferation and induced G1 phase arrest in EZH2-mutant DLBCL cell lines. The combination of EZH2 inhibitor SHR2554 with histone deacetylase (HDAC) inhibitor chidamide (hereafter referred to as HBI8000) exerted synergistic anti-proliferative activity in vitro and in vivo. Gene expression profile analysis revealed dramatic inhibition of the DNA replication process in combined treatment. Conclusions: SHR2554, a potent, highly selective small molecule inhibitor of EZH2, inhibited EZH2-mutant DLBCL more significantly in vitro and in vivo. The combination of HDAC inhibitor HBI8000 with EZH2 inhibitor SHR2554 exhibited dramatic anti-tumor activity in both mutant and wild-type DLBCL, which may become a potential therapeutic modality for the treatment of DLBCL patients.

## 1. Introduction

Diffuse large B-cell lymphoma (DLBCL) is the most common type of B-cell lymphoma, which can be divided into germinal center B-cell-like (GCB) and activated B-cell-like (ABC) subgroups according to gene expression profiling [1]. Although the prognosis of DLBCL patients has been improved by using anti-CD20 antibody rituximab in addition to chemotherapy, approximately 30–40% of DLBCL patients still develop resistance to this immunotherapy [2,3]. Thus, novel and effective therapeutic strategies are urgently needed for the treatment of recurrent or refractory DLBCL patients.

Histone methyltransferase EZH2 is the catalytic subunit of the Polycomb Repressive Complex 2 (PRC2), which is responsible for mono-, di- and tri-methylation of histone H3 lysine 27 (H3K27) and repression of gene expression [4,5]. EZH2 overexpression and somatic heterozygous mutations are implicated in dysregulation of histone modification and lymphomagenesis [6]. EZH2 hyperactivity and deregulated H3K27me3 are also correlated with poor prognosis in several cancer subtypes [7,8,9]. The functional mutations in EZH2 occur frequently in both GCB-DLBCL and follicular lymphoma (FL), which downregulate tumor suppressor genes and promote the proliferation of tumor cells [10]. Recurrent mutations of Tyr641 in EZH2 alter catalytic activity of PRC2 and induce increased levels of H3K27me3, which indicate that pharmacological inhibition of EZH2 activity may provide a novel therapeutic target for EZH2 mutant lymphoma [11,12]. Recently, several EZH2 inhibitors have exhibited promising therapeutic effects in GC-derived B-cell lymphoma patients bearing EZH2-activating mutations [13,14]. However, there are still many EZH2 wild-type DLBCL and FL patients with an objective response rate lower by 20% after treatment using EZH2 specific inhibitors. In this study, we present a novel highly selective EZH2 inhibitor SHR2554, which specifically inhibits both wild-type and mutant EZH2 methyltransferase activity with similar potencies and is currently undergoing clinical trials for the treatment of lymphoma patients (NCT03603951).

The epigenetic processes, especially histones acetylation, regulate gene expression through modification of chromatin structure and promotion of the access of related transcription factors to the DNA template, which also plays a crucial role for cancer development and tumorigenesis [15]. The dynamic balance between histones acetylation and deacetylation process is controlled by histone acetyltransferases (HATs) and histone deacetylases (HDACs) [16]. The aberrant activity of HDACs is frequently implicated in several lymphoid malignancies [17]. More importantly, two highly related histone and non-histone acetyltransferases, CREBBP/EP300 mutation, are detected in 39% of DLBCL and 41% of FL patients. The related somatic mutation induced cellular HAT reduction and decreased p53 tumor suppressor activity [18]. The presence of these genomic mutation and HAT defects indicated the therapeutic implications of HDAC inhibitors for the treatment of DLBCL patients. Recently, many reports have demonstrated that combination therapy with HDAC inhibitors improved the clinical benefit in the lymphoma patients [19,20]. In this study, we present a novel, highly selective EZH2 inhibitor SHR2554 and explore a possible combination strategy in DLBCL.

## 2. Materials and Methods

### 2.1. Drugs and Reagents

EZH2 inhibitor SHR2554 was provided by Jiangsu Hengrui Medicine Co., Ltd. (Jiangsu, China). Chidamide (CS055/HBI-8000) was kindly supplied by Chipscreen Biosciences Ltd. (Shenzhen, China). For in vitro experiments, the two compounds were dissolved in dimethylsulfoxide (Sigma–Aldrich, Darmstadt, Germany) at a concentration of 10 mM and stored at −80 °C. For in vivo experiments, they were dissolved in 0.4% carboxymethylcellulose sodium and 0.1% Tween-80 and stored at 4 °C. Antibodies against Caspase-3 (#9662S), Mcl-1 (#5453S), Bcl-xl (#2764S), XIAP (#2045S), CDK2 (#2546S), CDK4 (#12790S), CDK6(#13331S), H3K27me3(#9733T), H3K27ac(#9649S), H3(#4499S), EZH2(#5246S), P21(#2947S), PARP (#9532S, #5625S), ORC1 (#4371) were purchased from Cell Signaling Technology (Danvers, MA, USA). Anti-β-actin (Cat No. A5441) was purchased from Sigma (St. Louis, MO, USA). Anti-Ki67 (#ab16667) was purchased from Abcam (Cambridge, MA, USA).

### 2.2. Cell Lines and Culture Conditions

HBL-1, TMD8, OCI-LY7 and SU-DHL-16 cell lines were gifts from Dr. Fu, University of Nebraska Medical Center (Omaha, NE, USA). SU-DHL-2, SU-DHL-6, Pfeiffer, OCI-LY10, OCI-LY8, KARPAS-422 and U2932 cell lines were obtained from ATCC (Manassas, VA, USA) and DSMZ (Braunschweig, Germany). Cells were cultured in IMDM, DMEM or RPMI 1640 medium (Gibco, Life Technologies, Carlsbad, CA, USA) supplemented with 10–20% fetal bovine serum (FBS; Gibco, Life Technology) and penicillin-streptomycin in a humidified atmosphere of 5% CO_2_ at 37 °C.

### 2.3. Biochemical Assay

EZH2 and EZH2 mutants biochemical assay—EZH2 (WT, BPS, Cat. No. 51004), EZH2 (Y641C, Active Motif, Cat. No. 31389), EZH2 (Y641F, Active Motif, Cat. No. 31388), EZH2 (Y641N, Active Motif, Cat. No. 31390), EZH2 (Y641S, BPS, Cat. No. 51013) and EZH2 (A677G, Active Motif, Cat. No. 31391) were incubated with SHR2554 or EPZ-6438 at room temperature for 15 min. Then, 5 μL of substrate solution (Sigma, Cat. No. 7007) was added to each well and incubated for another 1 h at room temperature. The endpoint was read with EnSpire and IC_50_ values were calculated using GraphPad Prism (Version 8.4, San Diego, CA, USA). This study was conducted by ChemPartner.

H3K27 trimethylation assay—Pfeiffer cells were cultured in the absence or presence of SHR2554 or EPZ-6438 for 3 days. The concentrations of SHR2554 or EPZ-6438 were 1000, 200, 40, 8, 1.6, 0.32, 0.064, 0.0128 and 0.00256 nM, respectively. After incubation with the testing compound, H3K27 trimethylation in cells was examined with H3K27me3 Cellular Assay Kit (Cisbio, 62KC3PAE). IC_50_ value was calculated using GraphPad Prism by plotting the log(compound) concentrations versus percent inhibition of H3K27me3.

In vitro selectivity of methyltransferases—the selectivity of SHR2554 on 22 histone methyltransferases and 3 DNA methyltransferases was conducted by Eurofins Cerep. SHR2554 was tested at 0.01–100 µM. The IC_50_ values (concentration causing a half-maximal inhibition of control specific activity) and Hill coefficients (nH) were determined by non-linear regression analysis of the inhibition/concentration-response curves generated with mean replicate values using Hill equation curve fitting.

### 2.4. Cell Viability Assay

The exponentially growing cells were seeded in 96-well plates and the cell density depended on the cultured days. After treating with SHR2554 and/or HBI8000, cell viability was detected by Cell Titer-Glo Luminescent Cell viability assay system (Promega, Madison, WI, USA). Luminescent signals were measured by LMax II (Molecular Devices, Sunnyvale, CA, USA).

### 2.5. Apoptosis and Cell Cycle Assays

For apoptosis and cell cycle analysis, cells were treated with indicated concentrations of HBI8000 and SHR2554. Then, cells were collected and treated with Annexin V/PI apoptosis detection kit (BD Biosciences, San Jose, CA, USA) and PI staining buffer (Sigma–Aldrich, Darmstadt, Germany) assay system according to the manufacturer’s instructions. Finally, all samples were analyzed by BD Accuri C6 flow cytometer (BD, Biosciences, San Jose, CA, USA).

### 2.6. Western Blot and Real-Time Polymerase Chain Reaction (PCR)

Western blot and real-time PCR were performed as previously described [21]. The antibodies used were as above and the specific primers were as follows: ORC1 (forward primer: GTCCAATGTTGTAGCCGTGC, reverse primer: CGACGCTGAGATGGGATTGT) and GAPDH (forward primer: GCACCGTCAAGGCTGAGAAC, reverse primer: TGGTGAAGACGCCAGTGGA).

### 2.7. RNA-Seq

Cells were treated with the inhibitors at the indicated concentrations alone or in combination, then total RNA was purified by trizol method, and RNA integrity was confirmed by 2100 Bioanalyzer (Agilent Technologies Santa Clara, CA, USA). Sequencing was performed by HiSeq system (Illumina, San Diego, CA, USA) according to the manufacturer’s instructions, and data processing and analyzing were performed by Novogene Bioscience (Beijing, China).

### 2.8. Lentivirus-Mediated Small Hairpin RNA (lenti-shRNA) against ORC1

The Lenti-shRNA vector system (PGCSIL-GFP) was purchased and constructed from GeneChem Company (Shanghai, China). The ORC1 shRNA sequences were designed as follows: gcCACGTTTCAACAGATATAT, ccACCAAGTCTATGTGCAAAT. Non-silencing shRNA was used as the negative control vector.

### 2.9. In Vivo Experiments

The xenograft models, including two CDXs (SU-DHL6, U2932) and two PDXs (PDX001: EZH2 Y641N; PDX002: EZH2 WT), were constructed in this study. Non-obese diabetic/severe combined immunodeficient (NOD/SCID) mice (HFK Bioscience Co., Ltd. Beijing, China), aged 6–8 weeks, were used. For CDX models, tumor cells (6 × 10^6^) in 0.1 mL PBS medium with Matrigel (1:1 ratio) were injected subcutaneously into the area under the right flank of each mouse. Patient-derived lymphoma tissues were cut into fragments and then subcutaneously inoculated into 3–5 mice to construct the PDX models. When the tumor volume reached approximately 1 cm^3^, the mice were sacrificed, and tumor tissues were separated and reinoculated into new mice. When the tumor volume reached 100–150 mm^3^, mice were randomly divided into four groups: vehicle, HBI8000 (5 mg/kg, qd by gavage), SHR2554 (60 or 120 mg/kg, bid by gavage) and the combination. Tumor volume (V) and mouse weight (W) were monitored every three days, and the tumor volume was calculated using the following formula: V = (length × width^2^)/2. Tumor tissue samples were collected from all groups at 4 h after the last dose. All animal experiments were approved by the Institutional Animal Care and Use Committee of Peking University Cancer Hospital & Institute, and performed according to the guidelines for the care and use of laboratory animals.

### 2.10. Immunohistochemistry

The slides with 4 mm were incubated with primary antibody (Ki67: 1:200) overnight at 4 °C and then with HRP-conjugated secondary antibody at room temperature for 30 min. DAB was used for staining. The staining results were interpreted by two independent professional pathologists from the pathology department of Peking University Cancer Hospital in a double-blinded manner.

### 2.11. Statistical Analysis

Data were represented as mean ± SD from three independent experiments and representative results are shown in the figures. All statistical analyses were carried out using the IBM SPSS Statistics (Version 22.0; IBM Corp., New York, NY, USA). Data were analyzed using paired or unpaired Student’s *t*-test or one-way ANOVA and *p* < 0.05 was considered statistically significant. Combination index (CI) values < 1, =1 and >1 were used to indicate synergism effect, additive effect or antagonism effect, respectively, determined by the Chou–Talalay method using the CalcuSyn software (Version 2, Biosoft, Cambridge, UK).

## 3. Results

### 3.1. Biochemical Characterization of SHR2554 as a Potent and Selective Inhibitor of EZH2

EZH2 mediates tri-methylation (Me3) of lysine (K)27 on histone H3 (H3K27me3), which is responsible for the silencing of tumor suppressor genes in cancer cells and is purported to play a causal role in malignancies. Targeting EZH2 therapy has become a hot research topic in cancer treatment and here we present a potent and selective EZH2 inhibitor, SHR2554. A series of vitro studies was conducted to identify the potency and selectivity of SHR2554 on EZH2 and the first in class compound EPZ-6438 was also evaluated as a reference compound in these studies. SHR2554 and EPZ-6438 showed comparable inhibitory effect against wild-type and mutant EZH2 with IC_50_ ranging from 0.87 to 16.80 nM (Table 1). As EZH2 is responsible for H3K27 hyper-tri-methylation, the reduction in intracellular H3K27me3 levels following SHR2554 treatment was also examined in Pfeiffer lymphoma cell line. As shown in Table 2, both SHR2554 and EPZ-6438 could significantly reduce intracellular H3K27me3 levels in Pfeiffer cells. The IC_50_ values were 1.63 ± 0.14 and 4.13 ± 0.59 nM, respectively. Next, 22 histone methyltransferases and 3 DNA methyltransferases were tested to investigate the selectivity of SHR2554 on EZH2. SHR2554 was highly selective for EZH2 over most other methyltransferases tested, with selectivity greater than 10,000-fold. The only exception was EZH1, which is homologous to EZH2, with IC_50_ of 19.10 nM (22-fold over EZH2) (Table 3). These in vitro properties of SHR2554 showed that it is a highly potent and selective inhibitor of EZH2.

### 3.2. SHR2554 Inhibited Proliferation, Induced G1 Phase Arrest and Promoted Apoptosis in DLBCL Cell Lines

To explore the anti-tumor effect of SHR2554, the proliferation of four DLBCL cell lines (EZH2 MT: SU-DHL-6, KARPAS-422; EZH2 WT: U2932, SU-DHL-16) was first analyzed using the cell viability assay. As shown in Appendix A, SHR2554 showed anti-proliferative activity in a time- and dose-dependent manner. With the prolongation of drug treatment time, EZH2-mutant cell lines SU-DHL-6 and KARPAS-422 exhibited more sensitivity to SHR2554 treatment as compared to wild-type cell lines, which was consistent with previous reports that EZH2 inhibitors had a common feature of delay inhibition in EZH2-mutant cell lines [22]. According to these results, we selected 6 days of SHR2554 treatment for the other 7 cell lines. EZH2 mutant cell lines Pfeiffer, KARPAS-422 and SU-DHL-6 were more sensitive to SHR2554, with IC_50_ values less than 300 nM, while EZH2 wild-type cell lines were relatively resistant to SHR2554, with IC_50_ values greater than 600 nM except for SU-DHL-2 cells (Figure 1a). Meanwhile, we detected the basal expression of EZH2 in 11 cell lines to determine the relationship between EZH2 expression and drug sensitivity. The Western blot analysis indicated that EZH2 was highly expressed in almost all cell lines and there was no relationship between basal EZH2 expression and drug sensitivity (Appendix A).

To investigate the mechanisms by which EZH2 inhibitor induced cytotoxic effects, cell cycle and apoptosis were next analyzed by flow cytometry in U2932 and SU-DHL-6 cells. Cell cycle analysis showed that the number of cells in G1 phase significantly increased from 38.8% ± 1.5% to 62.0% ± 0.50% in SU-DHL-6 cells and from 68.1% ± 1.2% to 77.6% ± 1.1% in U2932 cells. The cell number in S and G2 phase reduced correspondingly after SHR2554 administration (Figure 1b). G1/S transition-related proteins (CDK2, CDK4, CDK6) were significantly decreased in SU-DHL-6 cells but slightly decreased in U2932 cells (Figure 1c). Similarly, consistent with the results of cell viability analysis, apoptotic cells increased dramatically from 14.2% to 50.9% in SU-DHL-6 cells but slightly increased from 3.1% to 7.9% in U2932 cells (Figure 1d). The pro-apoptotic protein cleaved-PARP and cleaved-Caspase-3 increased and the anti-apoptotic protein XIAP and MCL-1 decreased more significantly in SU-DHL-6 cell line compared with U2932 cells (Figure 1e). Taken together, these results indicate that SHR2554 inhibited proliferation more significantly in EZH2 mutant DLBCL cell lines than wild-type.

### 3.3. Synergistic Effect of EZH2 Inhibitor SHR2554 and HDAC Inhibitor HBI8000 on Induction of Cell Death in DLBCL Cell Lines

In view of the limitations of single drug, combined drug therapy has become the current trend of cancer treatment. To improve the anti-tumor efficacy of SHR2554 in EZH2 wild-type cell lines, synergistic anti-tumor activity of HDAC inhibitor HBI8000 and EZH2 inhibitor SHR2554 was next explored in DLBCL cell lines, especially for those without EZH2 mutation. Five DLBCL cell lines (EZH2 WT: SU-DHL-16, HBL-1 and U2932; EZH2 MT: KARPAS-422 and SU-DHL-6) were treated with indicated concentrations of SHR2554 and HBI8000 for 72 h. The concentrations were chosen according to 72 h IC_50_ per agent and per cell line, and the combination group was treated with SHR2554 and HBI8000 at a fixed ratio approximating their individual IC_50_. The effect of inducing cell death was assessed by calculating inhibition rate of cell proliferation. Combination of SHR2554 and HBI8000 intriguingly exerted higher growth inhibition than the single-agent group (Figure 2a). The extent of synergism was assessed by CI value. As shown in Figure 2b, the combination treatment induced a strong synergistic inhibition effect in SU-DHL-6 and U2932, with CI values ranging from 0.11 to 0.67, and triggered a medium synergistic inhibition effect in KARPAS-422 and HBL-1 cells, with CI values ranging from 0.6 to 0.89. Sequential drug administration with pre-treatment of SHR2554 for 72 h and co-treatment of SHR2554 and HBI8000 for an additional 72 h also demonstrated synergistic effect (Figure 2c,d). Overall, combination SHR2554 with HBI8000 interacted synergistically to inhibit cell growth in both EZH2 mutant and wild-type DLBCL cell lines.

### 3.4. Co-Treatment of SHR2554 and HBI8000 Induced Apoptosis, Cell Cycle Arrest in the G1/S Phase and Change of Histone Modification

To determine the status of cell cycle arrest and apoptosis after combination treatment, five DLBCL cell lines (EZH2 WT: SU-DHL-16, HBL-1 and U2932; EZH2 MT: KARPAS-422 and SU-DHL-6) were analyzed by flow cytometry and Western blot. Cells were first treated with indicated concentrations of SHR2554 and/or HBI8000 for 48 h. Flow cytometry analysis of SU-DHL-6 cells showed that the number of cells in G1 phase significantly increased from 32.3% ± 0.6% in the vehicle group to 47.2% ± 3.5% in the combination treatment group. Similar results were observed in KARPAS-422, SU-DHL-16, HBL-1 and U2932 cells (Figure 3a). Consistent with these results, the expressions of G1/S transition-related proteins (CDK2, CDK4, CDK6) decreased and the negative cell cycle regulator p21 increased substantially in the combination group (Figure 3b). On the other hand, co-treatment with indicated concentrations of SHR2554 and HBI8000 in SU-DHL-6 cells led to a significant increase in the percentage of apoptotic cells (from 7.9% ± 2.1% to 49.7% ± 8.9%). Similar results were observed in KARPAS-422, SU-DHL-16, HBL-1 and U2932 cells (Figure 3c). Similarly, the pro-apoptotic protein of cleaved-PARP and cleaved-caspase3 increased and the anti-apoptotic protein of XIAP, MCL1, Bcl-xL decreased significantly in combination group as compared to treatment with each agent alone (Figure 3d). Taken together, these results demonstrate that the combination of SHR2554 and HBI8000 could synergistically induce G1 phase arrest and apoptosis in both EZH2 mutant and wild-type DLBCL cell lines.

To further assess the effect of drug combination on acetylation and methylation of histone, H3K27ac and H3K27me3 were analyzed by Western blot. As shown in Figure 3e, treatment with SHR2554 alone dramatically reduced tri-methylation and elevated acetylation of H3K27. However, HDAC inhibitor HBI8000 not only increased acetylation of H3K27 but also elevated methylation of H3K27, which may explain the limited clinical effect of HDAC inhibitors to some extent. Importantly, the combination treatment led to a further elevation in H3K27 acetylation as compared to each agent alone and the elevated H3K27 tri-methylation caused by HBI8000 was attenuated by combined SHR2554, which may eventually lead to a more open chromatin structure and synergistic anti-tumor effect.

### 3.5. Gene Expression Signatures in DLBCL Were Affected by Combination Treatment of SHR2554 and HBI8000

To further investigate the mechanisms by which the SHR2554 and HBI8000 combination treatment induced the synergistic anti-tumor effect in DLBCL, gene expression profile was carried out after combination treatment. Compared with vehicle group, Venn diagram illustrated upregulated and downregulated gene changes of SHR2554, HBI8000 and combination treatment group. The majority of the genetic transcript increased or decreased by SHR2554 or HBI8000 alone was involved in the combination treatment group. For example, approximately 92% of transcripts (975 genes) upregulated by HBI8000 and 76% of transcripts (87 genes) upregulated by SHR2554 were contained in the combination treatment in U2932 cells (>1.5-fold, *p* < 0.05) (Figure 4a). Moreover, the Venn diagram also revealed that the epigenetic combination treatment offered new opportunities beyond what a single inhibitor had achieved and acted synergistically to regulate distinct gene expression profile. For example, 37 and 129 transcripts were efficiently decreased by the combination treatment (<−1.5-fold, *p* < 0.05), which was not present in SHR2554 or HBI8000 treatment alone in DLBCL cells (Figure 4b). More importantly, gene ontology (GO) enrichment analyses were further performed within these distinct gene expression profiles. The combination treatment might trigger synergistic anti-proliferative activity in both EZH2 wild-type and mutant tumors through downregulation of DNA replication and NIK/NF−kappaB signaling pathway and upregulation of negative regulation of MAPK cascade and B-cell activation (Figure 4a,b).

The gene set enrichment analysis (GSEA) was next performed based on RNA-Seq data to further investigate the mechanism underlying the related pathway regulated by combined treatment of EZH2 and HDAC inhibitor. Significant enrichment of DNA replication signature was observed in both EZH2 wild-type and mutant tumor cells, which was consistent with the results of cell proliferation and cell cycle in Figure 2 and Figure 3. Moreover, genes downregulated by combination treatment also showed significant enrichment of gene set in B-cell receptor signaling pathway (Figure 4c). Since the GO and GSEA analysis illustrated that the DNA replication process-related ORC1 expression was decreased in both U2932 and SU-DHL-6 cells, the mRNA expression of ORC1 was next analyzed by real-time PCR. As shown in Figure 4d, the results from qPCR analysis indicated that the combination treatment significantly decreased ORC1 expression in both EZH2 wild-type and mutant tumor cells. To further illustrate the possible anti-tumor mechanisms of combination treatment, the ORC1 shRNA was transfected into both U2932 and SU-DHL-6 cells. The knockdown of ORC1 expression suppressed proliferation of tumor cells, which indicated that ORC1 expression is critical to the survival of DLBCL cells (Figure 4e,f). Interestingly, a similar proliferation inhibitory effect was observed in ORC1 shRNA and combination treatment, which indicates that ORC1 expression was essential for DLBCL tumor cells and suppression of ORC1 in DNA replication process may contribute to the synergistic anti-tumor effect after co-administration of SHR2554 and HBI8000. 

### 3.6. Combination of SHR2554 and HBI8000 Exhibited Synergistic Anti-Tumor Effect in DLBCL Models In Vivo

Two cell-derived xenograft models (U2932: EZH2 WT; SU-DHL-6: EZH2 Y641N) and two patient-derived xenograft models (PDX001: EZH2 WT; PDX002: EZH2 Y641N) were used to assess the anti-tumor activity of drug combination in vivo. When the tumor volume was 100–150 mm^3^, the mice were randomized into four treatment groups according to tumor volume and body weight: vehicle, HBI8000 (5 mg/kg, qd), SHR2554 (60 mg/kg or 120 mg/kg, bid) and combination treatment. As shown in Figure 5a, SHR2554 exerted strong tumor suppressive activity in EZH2 mutant xenograft models SU-DHL-6 and PDX002, with TGI of 50% and 41% when 60 mg/kg agent was given by gavage twice a day. However, 120 mg/kg SHR2554 just induced moderate tumor suppressive activity in EZH2 wild-type models U2932 and PDX001, which was consistent with the previous results in vitro. Interestingly, the combination treatment exhibited a dramatic anti-tumor effect in EZH2 wild-type and mutant CDX and PDX xenograft models, demonstrating the potential synergistic anti-tumor effect of the two drugs in vivo. Tumor weights were also robustly restrained in combination groups (Figure 5b). More importantly, all treatments were well tolerated with no obvious body weight loss (Appendix A). To further elucidate the mechanism underlying tumor suppression, cell proliferation, cycle and apoptosis-related proteins were detected by Western blot and IHC. Consistent with previous results, the pro-apoptotic protein cleaved-PARP and negative cell cycle regulator p21 were significantly upregulated while Ki67 staining was significantly downregulated in combination groups as compared to monotherapy groups (Figure 5c,d). Therefore, these results revealed that combination of SHR2554 and HBI8000 had synergistic anti-tumor effect in DLBCL models in vivo.

## 4. Discussion

The EZH2-activating mutations are frequently observed in DLBCL and FL patients. The EZH2 Y641 mutation induced alteration of substrate preferences and enhanced methylation of H3K27me2 into H3K27me3. Although many EZH2 inhibitors were selectively designed for wild-type and mutant EZH2, most inhibitors just potentially inhibited the tumor-growth-bearing EZH2 mutations. Our study revealed for the first time that the combination of EZH2 and HDAC inhibitors produced marked anti-proliferative activity in both EZH2 wild-type and mutant status. Moreover, the combination treatment induced synergistic anti-tumor activity through the regulation of cell cycle, apoptosis and epigenetic-related protein. Based on this synergistic anti-tumor capacity, co-administration of EZH2 inhibitor and HDAC inhibitor could provide a potential therapeutic strategy for DLBCL patients.

The DNA methylation and histone modifications closely interacted and regulated the gene expression at both transcriptional and post-transcriptional levels. The simultaneous DNA demethylation and histone acetylation efficiently decreased proto-oncogenes expression, indicating that the inhibition of these processes could be a promising combination strategy for the treatment of cancer patients. Marchi et al demonstrated that the combination of hypomethylating agents and HDAC inhibitors exerted potential synergistic anti-tumor activity in preclinical models of T-cell lymphoma [23]. More importantly, a phase I clinical trial of the combination of DNA methyltransferase inhibitor decitabine and HDAC inhibitor vorinostat showed clinical activity with prolonged disease stabilization in advanced solid tumors and non-Hodgkin’s lymphomas [24]. Similarly, many reports demonstrated that the epigenetic disruption was also involved in pathogenesis and correlated with the clinical behavior of B-cell lymphoma [25]. Our study demonstrated for the first time that dual inhibition of methylation and deacetylation with SHR2554 and HBI8000 efficiently reduced the DLBCL tumor growth in vitro and in vivo. On the other hand, in multiple myeloma, acute myeloid leukemia/myelodysplastic syndromes (AML/MDS) and other high-risk hematological malignancies, the HDAC inhibitor has been used in combination with proteasome inhibitor bortezomib, anti-CD20 antibody rituximab and anti-CD22 antibody epratuzumab with promising synergistic activities and good tolerance [26,27,28].

During the investigation of biological mechanisms of this synergistic effect, the immunoblotting analysis showed that the combination treatment strongly induced the H3K27 acetylation, which indicated that the methylation modification may also alter the histone acetylation level in tumor cells. Eden et al. first demonstrated that DNA methylation also plays an important role in regulating the levels of chromatin acetylation [29], indicating that several DNA methyltransferases are associated with HDACs [30]. The methyltransferase Set7/9-catalyzed p53 methylation was closely related to the acetylation of p53 by acetyltransferase Tip60 [31]. More importantly, Wang et al. revealed that the knockout of EZH2 increased the acetylation level of H3K27 in brown preadipocytes [32]. On the other hand, the immunoblotting analysis also demonstrated that the elevated histone methylation level was accompanied with HDAC inhibitor treatment. Similarly, many recent reports demonstrated that several HDAC inhibitors modulated methylation profiles [33,34], which may result in resistance or side effects of HDAC inhibitors. Thus, the combination strategy of these epigenetic processes might be more promising and effective. 

In this study, gene expression profiling was also carried out to investigate the mechanisms of this synergistic anti-tumor activity using RNA-Seq analysis. Venn diagram illustrated that 129 transcripts were specifically reduced by the combination of epigenetic therapies. GO enrichment and GSEA analyses indicated that the inhibition of the DNA replication process was essential for these anti-lymphoma activities, indicating that the combination strategy targeted different selected profiles. Some reports indicated that DNA synthesis is precisely regulated by multiple genetic and epigenetic processes [35]. For example, Piunti et al. indicated that EZH2-knockout cells exhibited deficiency of DNA replication activity due to the absence of PRC2 activity [36]. Many HDACs, especially HDAC1 and 2, interacted with DNA synthesis factors and functioned in the DNA replication process [37]. More importantly, we showed that the combination of epigenetic treatments significantly downregulated the expression of DNA replication initiator protein ORC1 regardless of EZH2 mutation status of tumor cells. This notion indicated the great efficacy and synergistic anti-proliferative effects in tumor cells which might be attributed to the alternative selectivity profile provided by downregulation of ORC1 expression. However, further experiments are still needed to explore the precise mechanism between the synergistic effects of SHR2554 and HBI8000 and downregulation of ORC1 expression.

## 5. Conclusions

SHR2554, a potent, highly selective small-molecule inhibitor of EZH2, inhibited DLBCL with EZH2 mutation more evidently in vitro and in vivo. The combination of HDAC inhibitor HBI8000 and EZH2 inhibitor SHR2554 exhibited dramatic anti-tumor activity in both mutant and wild-type DLBCL, which could provide a potential therapeutic modality for the treatment of DLBCL patients.

## Figures and Tables

**Figure 1 cancers-13-04249-f001:**
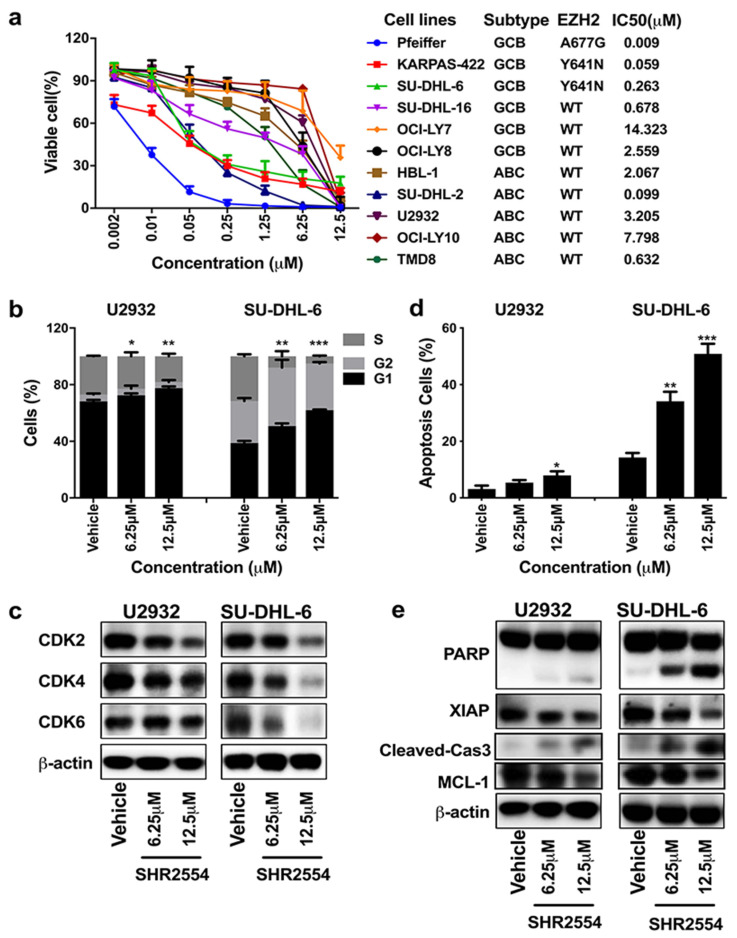
SHR2554 inhibited proliferation, induced G1 phase arrest and promoted apoptosis in DLBCL cell lines. (**a**) Eleven DLBCL cell lines were treated with indicated concentrations of SHR2554 for 6 days. Then the cell viability was measured by Cell Titer-Glo luminescent cell viability assay. Viable cells were calculated by dosing/vehicle × 100%. (**b**,**c**) Cells were treated with indicated concentrations of SHR2554 for 2 days. Then cell cycle was assessed by flow cytometry and cell-cycle-related proteins were detected by Western blot. (**d**,**e**) Cells were treated with indicated concentrations of SHR2554 for 6 days. Then apoptosis cells were assessed by flow cytometry and apoptosis-related proteins were detected by Western blot. Data are expressed as mean ± SD of three independent experiments and representative figures are presented. * *p* < 0.05, ** *p* < 0.01, *** *p* < 0.001, compared with vehicle group. Detailed information about Western Blot can be found at Appendix A.

**Figure 2 cancers-13-04249-f002:**
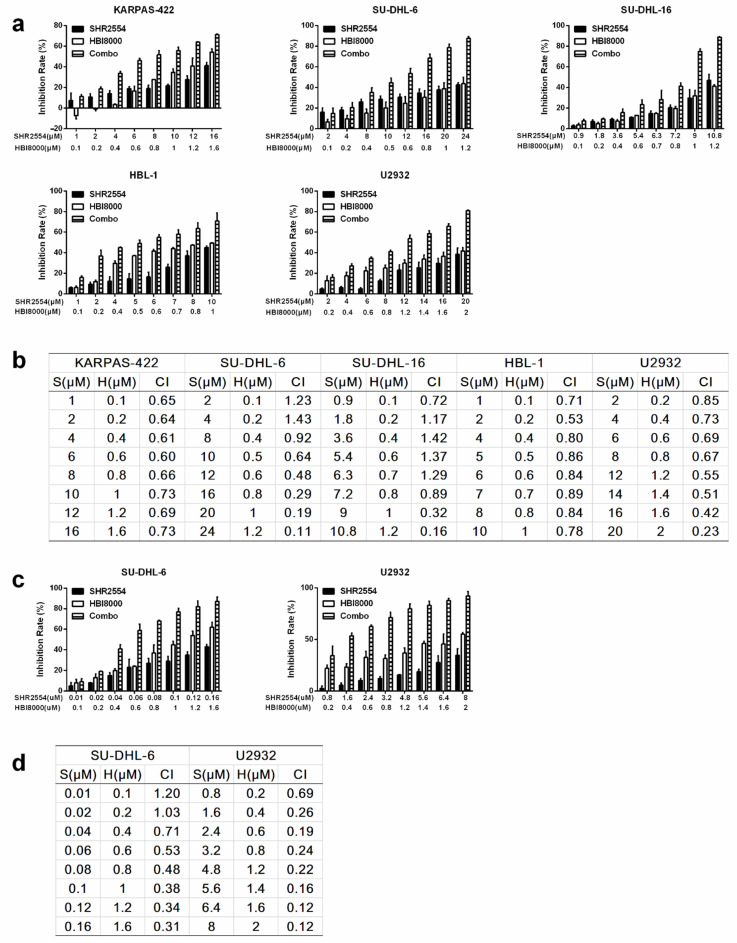
Synergistic effect of SHR2554 and HBI8000 on induction of cell death in DLBCL cell lines. (**a**) Concentrations of SHR2554 and HBI8000 were used for drug combination according to their 72 h IC_50_s in different cell lines and for each cell line, the ratio of SHR2554/HBI8000 was fixed. Cells were treated with indicated concentrations of SHR2554 or HBI8000 for 72 h and cell viability was measured by Cell Titer-Glo luminescent cell viability assay. Inhibition rates were calculated by (1−dosing/vehicle) × 100%. (**b**,**d**) Combination index was calculated by CalcuSyn software and CI values < 1 was considered to be synergistic. (**c**) Concentrations of SHR2554 and HBI8000 were used for drug combination according to their 144 h, 72 h IC_50_s, respectively, and the ratio of SHR2554/HBI8000 was fixed. Cells were treated with SHR2554 for 72 h first and co-treatment of SHR2554 and HBI8000 for an additional 72 h. Data are expressed as mean ± SD of three independent experiments. S: SHR2554; H: HBI8000; Combo: SHR2554 combined with HBI8000.

**Figure 3 cancers-13-04249-f003:**
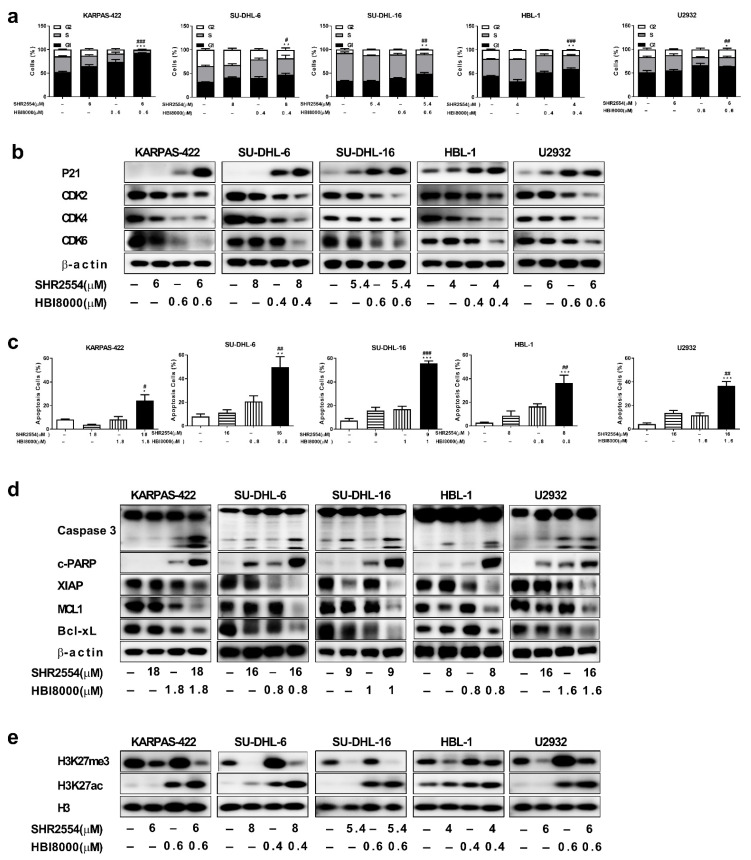
Co-treatment of SHR2554 and HBI8000 induces apoptosis, cell cycle arrest in the G1/S phase and change of histone modification. (**a**,**b**) Combination treatment induced G1 phase arrest in DLBCL cells. Cells were treated with indicated concentrations of SHR2554 and HBI8000 for 48 h. “–” indicated no inhibitor treatment. Then cell cycle was assessed by flow cytometry and cell-cycle-related proteins (CDK2, CDK4, CDK6, P21) were detected by Western blot. (**c**,**d**) Combination treatment prompted apoptosis in DLBCL cells. Cells were treated with indicated concentrations of SHR2554 and HBI8000 for 72 h. Then apoptosis determined by FITC+ PI− cells and FITC+ PI+ cells was assessed by flow cytometry and apoptosis-related proteins (Cleaved PARP, Caspase 3, XIAP, Mcl-1, Bcl-xL) were detected by Western blot. (**e**) Combination treatment increased acetylation of H3K27 in DLBCL cells. Cells were treated with indicated concentrations of SHR2554 and HBI8000 for 48 h. Then cells were collected and H3K27me3, H3K27ac were detected by Western blot. Representative figures are presented. Data are expressed as mean ± SD of three independent experiments and representative figures are presented. * *p* < 0.05, ** *p* < 0.01, *** *p* < 0.001, compared with vehicle group; # *p* < 0.05, ## *p* < 0.01, ### *p* < 0.001 compared with SHR2554 group. Detailed information about Western Blot can be found at Appendix A.

**Figure 4 cancers-13-04249-f004:**
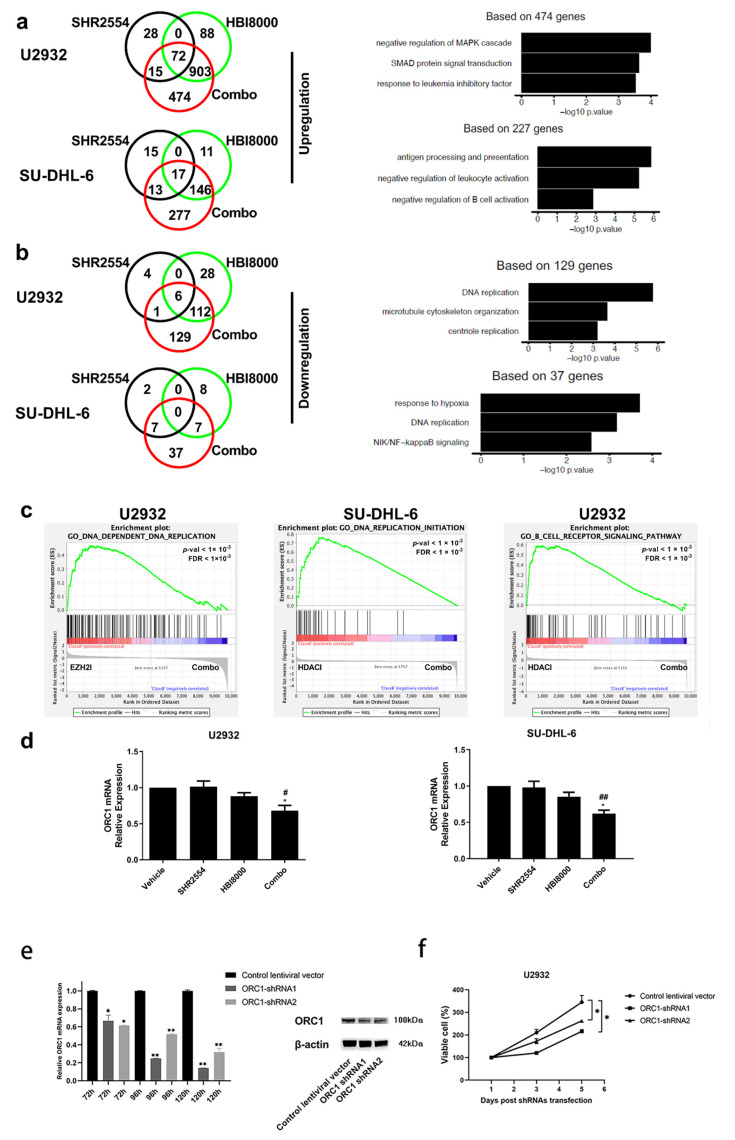
Gene expression signatures in DLBCL are affected by combination treatment of SHR2554 with HBI8000. U2932 and SU-DHL-6 cells were exposed for 48 h with SHR2554 (U2932: 16 μM, SU-DHL-6: 16 μM) and/or HBI8000 (U2932: 1.6 μM, SU-DHL-6: 0.8 μM). Then RNA was collected for sequencing. (**a**) Venn diagrams illustrating the number of the top upregulated gene changes. Based on these changed genes, top ranked pathways by GO analysis were represented. (**b**) Venn diagrams illustrating the number of the top downregulated gene changes. Based on these changed genes, top ranked pathways by GO enrichment analysis were represented. (**c**), Gene set enrichment (GSEA) plot depicting the enrichment of genes downregulated in DNA replication initiation (U2932 and SU-DHL-6) and B-cell receptor signaling pathway (U2932). (**d**) The mRNA expression of ORC1 gene in U2932 and SU-DHL-6 cells after being pre-treated with SHR2554 and/or HBI8000. Representative figures are presented. Data are expressed as mean ± SD of three independent experiments and representative figures are presented. * *p* < 0.05, ** *p* < 0.01, compared with HBI8000 group; # *p* < 0.05, ## *p* < 0.01, compared with SHR2554 group. (**e**) The U2932 cell was transfected with shRNA targeting ORC1, or treated with negative control lentiviral vector containing non-silencing shRNA. The expression of ORC1 in U2932 cell was detected using real-time PCR and Western blot. Detailed information about Western Blot can be found at Appendix A. (**f**) The cell viability of tumor cells was determined using the Cell-Glo luminescent cell viability assay after transfection. The results are represented of at least two similar experiments.

**Figure 5 cancers-13-04249-f005:**
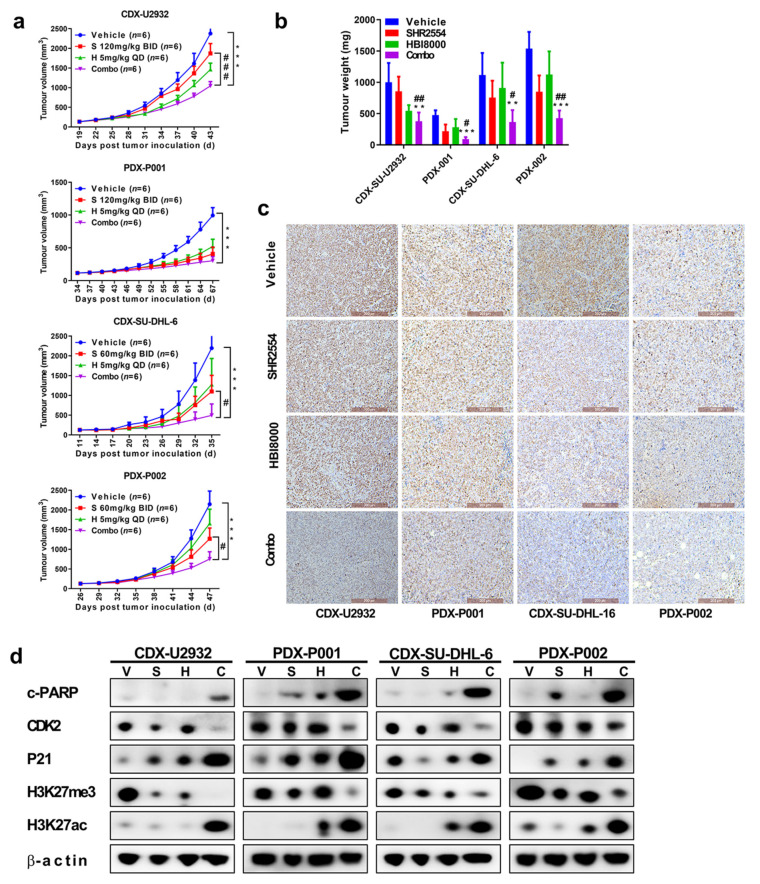
Combination of SHR2554 and HBI8000 exhibited synergistic anti-tumor effect in DLBCL models in vivo. Four DLBCL-derived xenograft models (CDXs: SU-DHL-6 and U2932; PDXs: PDX001-EZH2 WT and PDX002-EZH2 Y641N) were used to assess the anti-tumor activity of drug combination in vivo. (**a**) Tumor size curves derived from four models. (**b**) Tumor weight derived from four models. (**c**) Representative immunohistochemistry staining of Ki67 is shown (Scale bar 200 μM). (**d**) Tumor tissues from four models were used to evaluate cell cycle, apoptosis and histone modification-related pathway by Western blot. Detailed information about Western Blot can be found at Appendix A. ** *p* < 0.01, *** *p* < 0.001, # *p* < 0.05, ## *p* < 0.01, ### *p* < 0.001, compared with vehicle group.

**Table 1 cancers-13-04249-t001:** Potency of SHR2554 against wild-type and mutant EZH2.

Genes	IC_50_ (nM)	
	SHR2554	EPZ6438
EZH2	0.87 ± 0.02	0.96 ± 0.24
EZH2(Y641C)	16.80 ± 1.46	11.05 ± 1.43
EZH2(Y641F)	2.68 ± 0.41	2.44 ± 0.03
EZH2(Y641N)	1.79 ± 0.55	1.93 ± 0.23
EZH2(Y641S)	5.07 ± 0.24	4.45 ± 0.59
EZH2(Y641G)	1.13 ± 0.04	0.69 ± 0.18

**Table 2 cancers-13-04249-t002:** Effect of SHR2554 on H3K27me3 in Pfeiffer cells.

Compound	IC_50_ (nM)
	Mean ± SD
SHR2554	1.63 ± 0.14
EPZ-6438	4.13 ± 0.59

**Table 3 cancers-13-04249-t003:** Selectivity analysis of SHR2554 in a panel of methyltransferases.

Enzymes	Methyltransferases	Methylation Sites	IC_50_ (nM)	
			SHR2554	EPZ6438
Histone methyltransferases	EZH1	H3K27	19.10 ± 2.38	37.11 ± 15.72
	DOT1L	H3K79	>100 M	/
	EHMT1	H3K9	>100 M	/
	G9a	H3K9	>100 M	/
	MLL	H3K4	>100 M	/
	MLL2	H3K4	>100 M	/
	MLL3	H3K4	>100 M	/
	MLL4	H3K4	>100 M	/
	NSD3/WHSC1L1	H3K36	>100 M	/
	SET1B complex	H3K4	>100 M	/
	SETDB1	H3K9	>100 M	/
	SMYD2	H3K4/H3K36	>100 M	/
	SUV39H2	H3K9	>10 M	/
	SETD2	H3K26	>100 M	/
	SETD7	H3K4	>100 M	/
	SETD8	H4K20	>100 M	/
	PRDM9	H3K4/H3K36	>100 M	/
	PRMT1	H4R3	>100 M	/
	PRMT3	H4	>10 M	/
	PRMT4	H3R17/H3R26	>100 M	/
	PRMT5	H4R3	>100 M	/
DNA methyltransferases	DNMT1	-	>100 M	/
	DNMT3a	-	>100 M	/
	DNMT3b	-	>100 M	/

## Data Availability

The datasets used and analyzed during the current study are available from the corresponding author on reasonable request.

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
