# Peer review of "The Synergistic Anti-Tumor Activity of EZH2 Inhibitor SHR2554 and HDAC Inhibitor Chidamide through ORC1 Reduction of DNA Replication Process in Diffuse Large B Cell Lymphoma"

_cancers, 2021, doi:10.3390/cancers13174249_

Round 1
Reviewer 1 Report
The authors study the anti-tumor effects of a novel EZH2 inhibitor SHR2554 in diffuse large B-cell lymphoma. They demonstrate that SHR2554 is highly potent and selective for EZH2, and can significantly inhibit cell proliferation and tumor growth in EZH2-mutant DLBCL. They further show that SHR2554 and a HDAC inhibitor (chidamide) exert synergistic anti-tumor effects in both EZH2-mutant and EZH2 wild-type DLBCL. Gene expression profile analysis reveals dramatic inhibition of DNA replication process in the combined treatment. These findings have potential implications and provide insights into our understanding of lymphomagenesis.
Comment
- In the abstract, “Methods” is incomplete and should include other important experiments performed, such as cell viability, apoptosis, cell cycle, and etc.
- Please add Western blot findings for Caspase-3, MCL-1, Bcl-X1 in Figure 1e
- Figure 1c revealed a significant reduction of CDK6 in SU-DHL-6 cell line after treatment with 6.25uM SHR2554. However, the reduction of CDK6 was not significant in figure 3b when SU-DHL-6 cell line was treated with 8uM SHR2554.
- Some typos are noted. Please carefully proofread.
Reviewer 2 Report
The authors studied the effect of an EZH2 inhibitor on DLBCL and in addition in combination with an HDAC inhibitor. The study is extensive and sound. There are a few (minor) concerns:
- In simple summary: it says twice wide-type?? Should be wild-type?
- In figure 2 parts a/b and c/d seem very similar, could c and d be moved to the supplemental data?
- The gene expression data in figure 4: there is quite a difference in U2392 and SUDHL6 in gene expression in numbers but also in GO’s. Are there also genes that are changed that they have in common? Ofcourse this is only data from 2 individual cell lines, can there be much emphasis on such minimal results?
- Why was pcr done for ORC1, it is not clear why ORC1 was picked as a representative of what including shRNA knock downs. Please make clear why ORC1 was picked, based on which results!
Reviewer 3 Report
Wang et al revealed the synergistic effect of a novel EZH2 inhibitor and HDAC inhibitor in B-cell lymphomas. Firstly, this study described the effect of the drug combination in vitro in various cell lines exhibiting or not EZH2 mutattion. Second, using RNA-seq and biochemical assays, they deciphered the mechanism underlying this synergistic effect. Finally, they validated in vitro these observations, in animal models.
The manuscript is well written, the work is well conducted and the results are interesting and propose a novel therapeutic alternative for B lymphomas including DLBCL and FL which are still largely resistant to standard treatment.
1) It is not clear why the authors determined the IC50 of SHR2554 in the various cell line after 6 days of treatment (Fig 1a), while in the rest of the experiments the cells were treated for 72 hours
2) Could the authors discuss the high sensitivity of SUDHL2 to SHR2554 compared to the other EZH2WT cell lines ?
3) Figure 5 : Why no H3K27me3 increase was observed in vivo with HBI8000 ?
3) In vivo results have to be discussed.
Minor points :
Figure 3 and Figure 5c : too small !
Reviewer 4 Report
This manuscript, written by Dr. Xing Wang et al., original research, with the title of “The synergistic anti-tumor activity of EZH2 inhibitor SHR2554 2 and HDAC inhibitor chidamide through ORC1 reduction of 3 DNA replication process in diffuse large B cell lymphoma” focus on the effect on the cell cycle and apoptosis of diffuse large b-cell lymphoma (DLBCL) cells using the EZH2 inhibitor SHR2554 and the histone deacetylase inhibitor HBI-8000.
The authors performed elaborated in vitro and in vivo analysis with several DLBCL cell lines. The authors found that the combination of both drugs had an important anti-DLBCL effect in terms of cell cycle arrest (inhibition of proliferation) and induction of apoptosis.
This manuscript is well written, the English is correct, and contains much detailed information (that may overload the reader), the discussion, tables and figures look fine.
Comments:
1- In Table 1, can the authors please show that is the cut-off that was used to decide the performance of the inhibitor? IC50>100 um?
2- Why RNA-seq (NGS) technique was used to investigate the RNA expression? Which platform was used? Which genes were analyzed?
3- The authors used patient-derived lymphoma tissues to create the PDX models. Are there more information available about the characteristics of these DLBCL samples (clinicopathological features)?
4- In the GSEA figures. Which are the two phenotypes? Which are the nominal p values?
5- In the research of Figure 5. The authors performed immunohistochemistry for Ki67. The authors could have marked with other targets, such as apoptotic markers (CASPASE 3, or cleaved PARP), or EZH2. Nevertheless, it may not be feasible to do it now.
6- In Figure 3. I had difficulty understanding the “--” and the numbers such as “18”, “9”, etc…
7- The datasets, at least the gene expression, could be uploaded to GEO website.
